# Venetoclax in Relapse/Refractory AL Amyloidosis—A Multicenter International Retrospective Real-World Study

**DOI:** 10.3390/cancers15061710

**Published:** 2023-03-10

**Authors:** Eyal Lebel, Efstathios Kastritis, Giovanni Palladini, Paolo Milani, Foteini Theodorakakou, Shlomzion Aumann, Noa Lavi, Liat Shargian, Hila Magen, Yael Cohen, Moshe E. Gatt, Iuliana Vaxman

**Affiliations:** 1Faculty of Medicine, The Hebrew University of Jerusalem, Jerusalem 91120, Israel; 2Hematology Department, Hadassah-Hebrew University Medical Center, Jerusalem 91120, Israel; 3Department of Clinical Therapeutics, National and Kapodistrian University of Athens, GR 34400 Athens, Greece; 4Department of Molecular Medicine, University of Pavia, 27100 Pavia, Italy; 5Amyloidosis Research and Treatment Center, Fondazione IRCCS Policlinico San Matteo, 27100 Pavia, Italy; 6Department of Hematology, Rambam Health Care Campus, Haifa 3109601, Israel; 7Davidoff Cancer Center, Bellinson, 39 Jabutinsky Street, Petah Tikvah 4941492, Israel; 8Sackler Faculty of Medicine, Tel-Aviv University, Tel-Aviv 39040, Israel; 9Department of Hematology, Chaim Sheba Medical Center, Ramat-Gan 5265601, Israel; 10Department of Hematology, Tel-Aviv Sourasky Medical Center, Tel-Aviv 6423906, Israel

**Keywords:** venetoclax, amyloidosis, t(11;14)

## Abstract

**Simple Summary:**

Light-chain amyloidosis is a rare disease, and treatment for relapsed light-chain amyloidosis patients is an unmet need. Venetoclax is an oral medication that has proven great efficacy in many hematological cancers, including multiple myeloma. Venetoclax is considered a promising agent for the treatment of relapsed light-chain amyloidosis. We aimed to report the outcomes of therapy in 26 relapsed light-chain amyloidosis patients treated with venetoclax. We found a very high response rate (88%), and most responses were deep and prolonged. Treatment was effective even when doses were reduced. Venetoclax treatment was safe, and one patient died due to infection. These promising results require confirmation in a randomized trial.

**Abstract:**

Therapeutic options in relapsed refractory (R/R) light-chain (AL) amyloidosis patients are limited. Given the encouraging results in t(11;14) multiple myeloma and the high prevalence of t(11;14) in AL amyloidosis, venetoclax is an attractive treatment option in this setting. We report here the results of a multi-center retrospective study on 26 R/R AL amyloidosis patients treated off-label with venetoclax. The median lines of therapy prior to venetoclax was 3.5 (range 1–7), and 88% of our cohort had t (11;14). Twenty-two patients (85%) were previously treated with daratumumab. The overall hematologic response rate was 88%, 35% achieved a CR, and 35% achieved VGPR. The median event-free survival was 25 months (m) (95% CI 9.7 m-not reached), and the median overall survival was 33 m (95% CI 25.9–39.2 m). Most of the patients in this cohort are in ongoing deep responses and continuing venetoclax therapy. The treatment was relatively safe. One patient died due to infection, and there were two grade 3 infections in our cohort. Tumor lysis syndrome (TLS) was not seen in any patient. Dose reductions were frequent but did not affect the efficacy. These promising results require confirmation in a randomized controlled trial.

## 1. Introduction

AL amyloidosis (AL) is a rare plasma cell disorder characterized by progressive multi-organ damage due to misfolded light chains (LC) secreted by aberrant monoclonal plasma cells (PC) [1]. Thus, lowering LC levels to achieve affected organ responses and prolong survival has become the major goal of treatment. In the past two decades, novel agents have revolutionized the therapy and prognosis of all plasma cell disorders, including AL [1,2]. However, AL patients, as compared with multiple myeloma (MM) patients, are more frail and may not endure the adverse effects of the newer therapies. In addition, in some patients, the response to induction therapy is still suboptimal, while in others the disease eventually will relapse; hence, effective treatments for the second line and beyond are an unmet need [1,3,4]. Furthermore, promising therapies for advanced lines might play a role in the future as part of the initial treatment to obtain deep and prolonged LC reduction.

The selective BCL-2 inhibitor venetoclax has proven strong efficacy in many hematological cancers (ex., chronic lymphocytic lymphoma, acute myeloid leukemia, mantle cell lymphoma), which share its dependency on BCL-2 anti-apoptotic signaling [5,6,7]. In multiple myeloma (MM), certain molecular signatures predict response to venetoclax, especially t(11;14) and high expression of BCL-2 [8]. As t(11;14) is present in close to 50% of AL cases [9], venetoclax may become a promising therapy for this disease [10]. Moreover, multiple other mechanisms were recently shown to be involved in the sensitivity of AL clonal PC to the inhibition of BCL-2, and certain markers are being established to predict the response to venetoclax beyond t(11;14) [10].

Prospective studies’ data on venetoclax in AL are not available yet. A few single-center retrospective reports of the experience of venetoclax in up to twelve patients with AL were reported [11,12,13], showing promising results. A larger multicenter retrospective cohort with 43 patients was recently published, confirming its great efficacy and tolerability [14]. However, AL amyloidosis being a rare disease, data are scant and more studies to understand the role of venetoclax in AL are required.

In this study, we aim to add our experience of seven centers from three countries with venetoclax in relapsed/refractory (R/R) AL.

## 2. Methods

We retrospectively identified all AL amyloidosis patients who received at least one dose of venetoclax outside a clinical trial in all sites. The use of patients’ data for research purposes was approved by the local institutional review boards (IRB). Data were extracted from the medical records and included demographics, baseline disease characteristics, prior therapies, response to venetoclax, and adverse events on venetoclax therapy. The diagnosis and staging of amyloidosis were done according to consensus criteria [15]. Hematologic and organ responses were documented according to published criteria [16,17]. For the purpose of this study, hematologic responses were documented similarly, per AL amyloidosis criteria, in patients who had concurrent MM. Adverse effects were collected according to CTCAE. A patient was considered to have concurrent MM if he/she also had a myeloma-defining event. Hematologic progression was defined as an involved free light chain increase of 50% to >100 mg/L; a 50% increase in serum M-protein to >0.5 g/dL; a 50% increase in urine M-protein to >200 mg/day*;* or (in case of hematologic progression from complete response) a new abnormal free light chain ratio with doubling of the involved light chain [2]. Event-free survival (EFS) was defined as the time from the first day of venetoclax administration to hematologic progression; therapy change for inadequate response; or death. Overall survival (OS) was defined as the time from the first day of venetoclax administration to death from any cause. High-risk cytogenetics was defined as t(4;14), t(14;16), t(14;20), del17p, and 1q gain. The duration of response (DOR) was defined as the time from the first day of hematologic response to venetoclax to progression/death, in patients who achieved at least partial response.

Statistical analysis: categorical variables were described by numbers and percentages, and the difference between groups was evaluated using the chi-square test (for normally distributed variables) or by the Fischer exact test (for non-normally distributed variables). Continuous variables were described by mean and standard deviation and compared using a Student’s *T*-test (for normally distributed parameters) or by medians and range/interquartile range (IQR) and compared by the Wilcoxon signed-rank test (for non-normally distributed variables). The Kaplan–Meier method was used for DOR, EFS, and OS analysis, all statistical tests were two-sided, and *p*-values of <0.05 were significant. The median follow-up was calculated using the reverse Kaplan–Meier method. Statistical analysis was carried out using JMP 14 (SAS Institute, Cary, NC, USA) statistical software.

## 3. Results

Twenty-six AL patients from seven centers (15 from five different centers in Israel, 8 from Athens University, and 3 from Pavia Amyloid Center), who received venetoclax between March 2019 and October 2022, were identified and included in the study.

### 3.1. Baseline Characteristics

Table 1 shows the baseline characteristics of all patients at diagnosis. The median age at AL diagnosis was 64 years (range 50–86) and 15 patients (58%) were men. The median age at venetoclax initiation was 65 years (range 50–88) and 10 patients (38%) were aged 70 years or older. Eight patients (31%) had concurrent MM. Of 25/26 with available data on cytogenetics, 22 patients (88%) had t(11;14), and 4 (16%) had high-risk cytogenetics at diagnosis (all 4 had 1q gain and one had also del 17p). Most patients had cardiac and renal involvement (*n* = 20, 77%, and *n* = 15, 58%, respectively). The median number of organs involved was 2 (range 1–6). Fourteen patients (54%) had Eastern cooperative oncology group performance statuses (ECOG-PS) of ≥2 and 4 (15%) had ECOG-PS of 3–4.

The median number of prior lines of therapy was 3.5 (range 1–7). Five patients (19%) underwent prior autologous stem cell transplantation (ASCT). All patients (100%) previously received bortezomib, 22 patients (85%) previously received daratumumab, and 17 patients (65%) previously received lenalidomide. Other prior therapies are detailed in Table 1. The median time from diagnosis to venetoclax initiation was 12 months (m) (interquartile range (IQR) 5–41).

### 3.2. Treatment and Response

Table 2 shows the characteristics of and response to venetoclax treatment. Nine patients (35%) received venetoclax monotherapy and nine (35%) received venetoclax in combination with dexamethasone. Eight patients (31%) received venetoclax in combination with daratumumab (with or without dexamethasone), and one of them also with bortezomib (=venetoclax, daratumumab, bortezomib, and dexamethasone). All eight patients who received venetoclax with daratumumab were already receiving daratumumab, and, due to insufficient response, venetoclax was added to the regimen or replaced another agent (lenalidomide/pomalidomide) while daratumumab was continued.

The median venetoclax dose was 400 mg/day (d) (range 200–800 mg). At the time of venetoclax initiation, seven patients (27%) had creatinine over 2 mg/dL, all of whom received a dose of 400 mg/d or higher.

The overall response rate (ORR) to venetoclax-based therapy (response was considered as at least partial hematologic response) was 23/26 (88%). Nine patients (35%) achieved a complete hematologic response (CR), nine patients (35%) achieved a very good partial hematologic response (VGPR), and five patients (19%) achieved a partial hematologic response (PR). Three patients (12%) had no response (NR). Thus, 18/26 (69%) achieved a depth of response of VGPR or CR. The median duration of response (mDOR, for the 23/26 patients with at least PR) was 25 months (95% CI 8–29 m). Within the subgroup of patients who achieved VGPR/CR, the mDOR was similar at 25 months (95% CI 2–29 m). Of the eight patients that received venetoclax in combination with daratumumab, three patients achieved CR, three patients VGPR, one patient PR, and one patient NR. Three patients in our cohort did not harbor t(11;14). One achieved CR, one PR, and one NR. The patient that did not respond is the only patient in our cohort that died due to toxicity (see below).

Detailed serial biomarker-based evaluation of organ responses was missing in many patients. Yet, in terms of organ responses meeting official biomarker-based criteria, five documented cardiac responses were observed, and an additional two patients were judged as having marked clinical improvement of heart failure. Four renal responses and two hepatic responses were also reported. One additional patient had an improvement in gastric involvement (abdominal pain and mucosal endoscopic appearance have markedly improved).

### 3.3. Survival Outcomes

The median follow-up from initiation of venetoclax therapy was 33 months (95% CI 26–39). At data cutoff, 20 patients (77%) were alive. Five patients (19%) died of disease complications (one patient within the first cycle, three patients within the first 6 months of therapy, one patient later than 6 months) and one (4%) died due to infection. All five patients died due to disease complications (three sudden cardiac death, one heart failure, one unknown).

The median EFS was 25 m (95% CI 9.7 m—not reached). The median OS for the whole cohort was 33 m (95% CI 25.9–39.2 m). Figure 1 shows the EFS and OS curves.

### 3.4. Toxicity

Table 3 summarizes the toxicities during venetoclax therapy. A total of nine grade 3–5 toxicities were documented in the study, in a total of five patients (19%): three of these were infections, two of grade 3 and one of grade 5, one diarrhea of grade 3, while the others were hematological toxicities. Twelve patients (46%) received antibiotic prophylaxis while on venetoclax. Six patients (23%) had infections, three were grade 1–2 and three grade 3–5. Tumor lysis syndrome (TLS) was not seen in any patient in our cohort.

Dose reductions occurred in 10 patients (38%), mostly due to hematological toxicity. Excluding one patient, all patients with dose reductions continued with the reduced dose. The mDOR of patients with/without dose reductions was similar. The median tolerated dose was 400 mg (IQR 200–500).

At last follow-up, twenty patients (77%) were still on therapy. The reasons for treatment discontinuation were hematologic PD in three patients and toxicity in three patients (one due to hematological toxicity and two due to infections).

## 4. Discussion

In this multi-center study, we report promising outcomes in a cohort of 26 AL patients treated with the BCL-2 inhibitor venetoclax outside of a clinical study, with an ORR of 88%, CR rate of 32%, mDOR of 25 m, median EFS of 25 m, and median OS of 33 m. At data cut-off, most of the patients in this cohort are in ongoing deep responses and continuing therapy with venetoclax. The promising efficacy of venetoclax reported here is especially encouraging, since this is a heavily pretreated cohort, having received a median of 3.5 (range 1–7) prior lines of therapy, including daratumumab and bortezomib. Table 4 compares the results of previous retrospective studies to the current cohort. In our study, the ORR was higher than most other reports, possibly due to the high proportion of t(11;14).

Detailed serial biomarker-based evaluation of organ responses was missing in many patients, partially because of short follow-up for some of the responders. Still, clinical benefit was definitely observed in many patients, as detailed above. As organ response tends to be delayed in AL, and many patients are still on therapy with an ongoing hematological response, we expect that, in the future, more patients will gain organ responses. That said, a delayed organ response that is related to previous therapies cannot be excluded and cannot be differentiated accurately from the benefit that is attributed to venetoclax.

The majority of patients in our cohort (88%) had t(11;14), a biomarker clearly associated with favorable efficacy in MM [18,19] and also in AL [14]. Though a recent pivotal molecular study by Frazer et al. [10] and other studies may lead to better molecular definitions, in the near future, the presence of t(11;14) in PC will continue to be an accessible, practical marker that predicts a favorable response to venetoclax. More data is needed to assess the value of venetoclax therapy for non-t(11:14) AL patients, which is definitely less compared with t(11:14) patients.

In our cohort, the outcomes of the 31% of patients receiving venetoclax in combination with daratumumab were not more favorable than those receiving single-agent venetoclax or venetoclax with dexamethasone. However, almost all of these patients had insufficient response or even progression while on daratumumab-based therapy before adding venetoclax. Additionally, the number of patients in each subgroup (venetoclax with/without daratumumab) was small. Hence, no conclusions can be drawn regarding the value of venetoclax–daratumumab combinations or any other venetoclax combination. Venetoclax showed remarkable efficacy when combined with carfilzomib R/R myeloma patients [20]. Moreover, as BCL-2 inhibitors were found in vitro to sensitize AL PC to bortezomib, dexamethasone, and lenalidomide [10], combinations of venetoclax and these agents and/or others might be of interest. However, toxicity may be a significant limit, especially in AL. The combination of venetoclax with bortezomib was found to be hazardous in MM [21], and carfilzomib is seldom utilized in AL patients due to its cardiotoxic properties.

Venetoclax was, overall, well tolerated in our cohort, including a sizeable proportion of frail patients, with 77% of patients with cardiac involvement and 39% with advanced Mayo-stage 3a and 3b. In addition, 38% had an ECOG-PS of 2 and 14% had ECOG-PS of 3–4. These patients are usually not included in clinical trials and endured treatment well. One fatal infection occurred during venetoclax therapy; however, the infection rate generally, and severe infections rate particularly, were relatively low. The toxicities seen in our study and other reports compare favorably with the toxicity observed in studies with venetoclax in MM [19]. We observed three (11%) grade 3–5 infections and four (15%) grade 3–4 cytopenias. In the BELLINI clinical trial [21], patients treated with venetoclax, bortezomib, and dexamethasone were found to have a higher rate of grade 3–5 infections of 30%. In a Phase 2 study that reported the outcomes of the 31 MM patients [22], 19% died (five due to PD, and one from an adverse event (AE)) and three patients (10%) developed sepsis. In our cohort, AEs of grade ≥ 3 occurred in six patients, which were limited to cytopenias or infections, and led to discontinuation of therapy in three of the six patients.

However, cytopenias were a major cause of dose reduction in our cohort of patients. The majority were treated with a dose of 400 mg QD or less (from initiation of therapy and/or following a dose reduction). In the largest study, including 43 AL patients, detailed dosing data were not reported (other than a range of 100–800 mg) [14]. In a single-center report of 10 AL patients, all patients were treated with a venetoclax daily dose of 100–400 mg [12], and in a report of another single-center cohort of 12 patients [11] they were treated with doses of 400 or 800 mg of venetoclax daily, yet dose reductions were not reported. Interestingly, in our study, among the six patients who had grade ≥3 cytopenias/infections, only the patient with grade 5 infection was a non-responder, while all of the other five responded to therapy, three of them with CR. These data raise a question regarding the optimal dose needed to avoid AE on one hand but retain effectiveness on the other hand, especially in the complicated population of AL patients. Dose reductions in our study seemed to have no effect on the DOR in the deeply responding patients. We hypothesize that the small PC clones in AL may require lower doses. However, this question can only be answered in a prospective clinical trial.

As expected, but still relevant to this agent, tumor lysis syndromes were not observed. Gastrointestinal toxicity, which can be significant with venetoclax, and of interest in AL due to pre-existing symptoms in some patients, was not significant in our study. Importantly, cardiac toxicity and decompensation were not noted, making venetoclax therapy an attractive compound for AL patients.

This study has several limitations. This is a retrospective report, follow-up was not standardized, and, therefore, some data were missing, mostly regarding organ responses. The sample size was small (*n* = 26) due to the rarity of this disease and the fact that venetoclax is not approved and not commercially available either for AL or for MM.

## 5. Conclusions

In this study, we summarized our experience with venetoclax in 26 AL patients from seven centers in Israel/Greece/Italy. Our results support those of smaller single-center reports and a larger US/European study, showing encouraging effectiveness and a good safety and tolerability profile for this oral agent. Given the biological rationale and the accumulating data supporting its usage, we believe that prospective studies with venetoclax in various settings, including earlier in the AL course, as well as in various combinations, should be accelerated and encouraged.

## Figures and Tables

**Figure 1 cancers-15-01710-f001:**
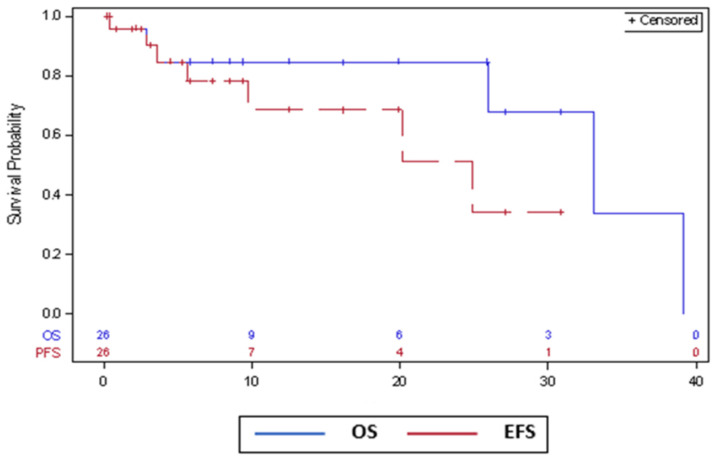
Event-free survival (EFS) and overall survival (OS) curves.

**Table 1 cancers-15-01710-t001:** Characteristics of the 26 AL amyloidosis patients included in the study.

Variable		Cohort (*n* = 26)
Age at venetoclax initiation, median (range) years		65 (50–88)
Males (*n*)/Females (*n*)		15/11
Median bone marrow plasma cells at diagnosis, % (IQR)		20 (13–30)
Involved light chain- Kappa, *n* (%)		9 (35)
Lambda, *n* (%)		17 (65)
Median dFLC at diagnosis, mg/L (IQR)		348 (120–551)
Involved organs—*n* (%)	Heart	20 (77)
	Kidneys	15 (58)
	PNS	6 (23)
	GI	7 (27)
	Soft tissue	10 (38)
	Liver	4 (15)
Cardiac stage—*n* (%)	1	6 (23)
	2	9 (35)
	3a	8 (31)
	3b	2 (8)
	missing	1 (4)
t(11;14) translocation— *n* (%)		22/25 (88)
Concurrent clinical MM— *n* (%)		8 (31)
Performance status at venetoclax initiation *, *n* (%)		
0–1		12 (46)
2		10 (38)
3–4		4 (15)
No. of prior lines of therapy, median (range)		3.5 (1–7)
Prior therapies **—*n* (%)	Bortezomib	26 (100)
	Lenalidomide	17 (65)
	Pomalidomide	13 (50)
	Daratumumab	22 (85)
	Alkylators	23 (88)
	ASCT	5 (19)
Time from Diagnosis to venetoclax, month (IQR)		12 (5–41)
Hemoglobin at venetoclax initiation, g/dL (IQR)		11.6 (10.6–12.1)
Platelets at venetoclax initiation, ×10^9^/L (IQR)		198 (166–235)
ANC at venetoclax initiation, ×10^9^/L (IQR)		4.1 (2.2–6.5)
Creatinine at venetoclax initiation, mg/dL (IQR)		1.13 (0.8–2)

As all percentages were rounded, the sum of percentages for each variable is not always 100%. Abbreviations: IQR—interquartile range; K—kappa; L—lambda; t—translocation; dFLC—difference between involved and uninvolved free light chains; PNS—peripheral nervous system; GI—gastrointestinal system; MM—multiple myeloma; ASCT—autologous stem cell transplant. * By Eastern cooperative oncology group (ECOG) scale. ** Other therapies included ixazomib (3 patients), thalidomide (2), elotuzumab (2), carfilzomib (1), belantamab (1), rituximab (1) and NEOD001 (1).

**Table 2 cancers-15-01710-t002:** Venetoclax treatment—characteristics and response.

Variable		Cohort (*n* = 26)
Venetoclax maximum daily dose *—median (range)		400 mg (200–800)
Venetoclax combination—*n* (%)	single agent	9 (35)
	with DEX	9 (35)
	with DARA (±DEX)	7 (27)
	with DARA + BOR + DEX	1 (4)
Overall response rate—*n* (%)		23 (88)
Quality of response, *n* (%)	CR	9 (35)
	VGPR	9 (35)
	PR	5 (19)
	NR	3 (12)
Time to any response—median (range) months		1 (0.3–12)
Time to best response—median (range) months		2 (0.3–11)

As all percentages were rounded, the sum of percentages for each variable is not always 100% Abbreviations: DEX—dexamethasone; DARA—daratumumab; BOR—bortezomib; CR—complete response; VGPR—very good partial response; PR—partial response; NR—no response. * Considering drug interactions (azoles and other drugs).

**Table 3 cancers-15-01710-t003:** Toxicities during venetoclax therapy.

	Infections	Hematological Toxicities	TLS	GI Toxicities	Dose Reductions	Treatment Discontinuation
Venetoclax monotherapy	1 G2	1 G4 TCP 2 G1-2 anemia 1 G1 neutropenia	0	0	2 patients	two discontinued due to PD
Venetoclax + DEX	2 G3 infections, 1 G5 infection	1 G1 TCP 1 G3 TCP 1 G3 anemia 1 G2 anemia 1 G4 neutropenia 1 G1 neutropenia	0	1 G1 diarrhea; 1 G3 diarrhea	4 patients	two discontinued due to toxicity
Venetoclax + DARA) ±DEX, +BOR in 1 patient)	2 G1-2	3 G1-2 TCP 2 G1 anemia 1 G3 neutropenia 2 G1-2 neutropenia	0	0	4 patients	two discontinued due to PD and one due to toxicity

Abbreviations: G—grade; TCP—thromvocytopenia; TLS—tumor lysis syndrome; GI—gastrointestinal; DEX—dexamethasone; DARA—daratumumab; BOR—bortezomib; PD—progressive disease.

**Table 4 cancers-15-01710-t004:** Studies reporting on venetoclax in AL amyloidosis.

	Sidiqi 2020 BCJ [11]	Pasquer 2021 BJH [12]	Nahi * 2021 AJH [13]	Premkumar 2021 BCJ [14]	Current Cohort
Number of patients	12	10	8	43	26
% t(11;14)	92%	70%	100%	72%	88%
Median prior lines	2 (range 1–4)	Not reported (70% 3 + pervious lines)	Not reported	3	3.5 (range 1–7)
Daily doses	7–800 mg; 5–400 mg	5–400 mg; 4–200 mg; 1–100 mg	400 mg	100–800 mg	Median 400 mg, range 200–800
ORR %	88%	66.6%	71%	68%	88%
Infections	in 2 patients	Not reported	Not reported	7% grade 3+	11% G3-5
TLS	0	0	0	0	0
G3+ cytopenias	Not reported	1 patient (10%) with anemia and grade 3 thrombocytopenia	Not reported	9%	11% G3-4
Treatment discontinuation due to toxicity	16%	30%	Not reported	19%	8%
Death on therapy	0	5 patients (50%) died: 3 from heart failure not attributed to venetoclax, 1 from infection and 1 from an unknown cause	0	1 patient died due to sepsis and 1 due to heart failure not attributed to venetoclax	1 patient died due to infection
mDOR	Not reported	241 days	Not reported	Not reported	25 months
mPFS	Not reported	Not reported	Not reported	31 months ‡	25 months ‡
mOS	Not reported	10.5 months	Not reported	Not reached	33 months

Abbreviations: ORR—overall response rate; G—grade; TLS—tumor lysis syndrome; mDOR—median duration of response; mPFS—median progression-free survival; mOS—median overall survival. * This study reported on t(11:14) MM and AL patients. Some of the data in the table are missing, as the study did not report on all variables in AL patients separately. ‡ In Premkumar et al. [14], progression-free survival was reported; in the current study, event-free survival is reported (capturing hematological progression/change in therapy for inadequate response/death as events).

## Data Availability

Data can be shared up on request.

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
