# Peer review of "Venetoclax in Relapse/Refractory AL Amyloidosis—A Multicenter International Retrospective Real-World Study"

_cancers, 2023, doi:10.3390/cancers15061710_

Round 1
Reviewer 1 Report
The authors studied retrospectively in a multicenter study the effect of Venetoclax treatment in 26 relapsed/refractory AL amyloidosis patients. The study confirms the favorable effects of Venetoclax in such patients obtained in similar smaller (and one bigger) retrospective studies. Most patients had t(11:14) translocation, median event-free survival was 25 months, median overall survival was 33 months and most patients had ongoing deep responses while continuing Venetoclax. Adverse effects were acceptable in this heavily pretreated group of patients.
Comments:
· This is a well-written manuscript, concise and harboring a lot of relevant information. Notwithstanding its retrospective nature, important parameters such as hematological responses, organ responses, overall survival, event free survival, duration of response, toxicity and infections were all presented.
· A disadvantage is the presence of multiple tiny discrepancies in the text and between the text and the tables. Though small, such inaccuracies unnecessarily harm the credibility of the manuscript and that is a pity for such a relevant study.
· To mention some of these:
o Line 101: 10 patients (38%).
o Line 102: Eight patients (31%).
o Line 102: Twenty-two patients (85%).
o Line 107: and 4 (15%).
o Table 1:
§ sometimes the IQR is represented by (xx, xx) and sometimes by (xx-xx). This is not consistent.
§ the correct figures of Age at Venetoclax initiation should be 65 (50-88) if identical to the text in line 101.
§ Kappa 9 (35)
§ Lambda 17 (65)
§ cardiac stage 2 9 (35)
§ Performance status 3-4 4 (15)
o Line 123: it is unclear to me why 34.5% is stated here instead of 35% and again four times in Table 2 and in lines 138 and 139 whereas all other percentages are presented as integers.
o Line 166-171: This is not completely clear to me. Five grade 3-5 toxicities during Venetoclax of which three infections leaves two hematological ones in my opinion. If I look at Table 3, I see in the second column four hematological grade 3-5 toxicities (later also mentioned in line 231). Please make it more clear for the reader in lines 166-171. And only three grade 3-5 infections were presented in the first column of Table 3. The other three grade 1-2 infections are not shown in the table whereas such low grade hematological or GI toxicities are.
o Line 175: in the text dose reductions occurred in 11 patients whereas in Table 3 only 10 can be found in column 5.
Author Response
The authors studied retrospectively in a multicenter study the effect of Venetoclax treatment in 26 relapsed/refractory AL amyloidosis patients. The study confirms the favorable effects of Venetoclax in such patients obtained in similar smaller (and one bigger) retrospective studies. Most patients had t(11:14) translocation, median event-free survival was 25 months, median overall survival was 33 months and most patients had ongoing deep responses while continuing Venetoclax. Adverse effects were acceptable in this heavily pretreated group of patients.
Comments:
- This is a well-written manuscript, concise and harboring a lot of relevant information. Notwithstanding its retrospective nature, important parameters such as hematological responses, organ responses, overall survival, event free survival, duration of response, toxicity and infections were all presented.
- A disadvantage is the presence of multiple tiny discrepancies in the text and between the text and the tables. Though small, such inaccuracies unnecessarily harm the credibility of the manuscript and that is a pity for such a relevant study.
- To mention some of these:
o Line 101: 10 patients (38%).
o Line 102: Eight patients (31%).
o Line 102: Twenty-two patients (85%).
o Line 107: and 4 (15%).
o Table 1:
- sometimes the IQR is represented by (xx, xx) and sometimes by (xx-xx). This is not consistent.
- the correct figures of Age at Venetoclax initiation should be 65 (50-88) if identical to the text in line 101.
- Kappa 9 (35)
- Lambda 17 (65)
- cardiac stage 2 9 (35)
- Performance status 3-4 4 (15)
We totally agree that the tiny discrepancies are inappropriate, and thank the reviewer for the meticulous review. We corrected all discrepancies. Some of the discrepancies stem from our wish that all percentages will sum to 100%. For consistency, we rounded now each percentage individually, and added a comment in the end of the table.
- Line 123: it is unclear to me why 34.5% is stated here instead of 35% and again four times in Table 2 and in lines 138 and 139 whereas all other percentages are presented as integers.
We agree, see above. For consistency, all percentages are now rounded, and presented as integers, even if the sum of percentages is not 100%.
- Line 166-171: This is not completely clear to me. Five grade 3-5 toxicities during Venetoclax of which three infections leaves two hematological ones in my opinion. If I look at Table 3, I see in the second column four hematological grade 3-5 toxicities (later also mentioned in line 231). Please make it more clear for the reader in lines 166-171. And only three grade 3-5 infections were presented in the first column of Table 3. The other three grade 1-2 infections are not shown in the table whereas such low grade hematological or GI toxicities are.
Thank you for this comment, the data is correct, but we totally agree that it can be presented more clearly. 5 patients (not 5 AE's), as mentioned, had G3-5 AE's, however, some patients had more than one G3-5 AE, hence the total G3-5 AE's were 9. We revised. Also, as suggested, we added all AE's, including G1-2 to the table, for consistency between different types of AE's.
- Line 175: in the text dose reductions occurred in 11 patients whereas in Table 3 only 10 can be found in column 5.
Thank you for this comment, the text was corrected, as there were indeed only 10 dose reductions.
Reviewer 2 Report
There is a paucity of data on effective treatments for relapsed patients with systemic AL amyloidosis, especially for patients refractory to daratumumab. In this multicenter retrospective cohort study, Lebel et al. report the clinical outcomes of 26 heavily pretreated patients with relapsed system AL amyloidosis treated with venetoclax. Given the rarity of the disease, this is a sizeable cohort, and this paper adds to the evidence that venetoclax appears to be effective and relatively safe. The manuscript is thorough and very well written.
Minor feedback:
- Line 78-79: Please clarify how hematologic response was determined in the 31% of patients with concurrent MM?
- Line 86: what was the definition of “inadequate response” used for EFS?
- Line 109: the median age at venetoclax initiation does not match the first row I the table, please clarify
- Table 1: Despite having 3.5 median prior lines of therapy, the median time from diagnosis to venetoclax initiation was 12 months, meaning that patients in this cohort had very short prior treatment durations – was this because of intolerance of lack of response? What was the best organ and hematologic response to prior therapy? (This will be helpful to determine if early organ responses to venetoclax may have been due to heme responses to prior therapy)
- Table 1: ANC at venetoclax initiation?
- Table 1: is the median prior therapies equivalent to the median prior lines?
- Table 1: what was the median dFLC at initiation of venetoclax
- Table 2: how can the lower range of time to any response and time to best response be zero? Wouldn’t that imply that patients responded on day zero of receiving venetoclax?
- Section 3.3: Of the 5 patients who died from disease complications, what were the causes of death?
- Line 168: In the EFS outcome, what proportion of events was due to organ/heme progression versus “inadequate response”
- Line 182 and Table 2: Table 2 shows that the median “effective dose” was 400mg/day, how was this defined (is this the median dose among patients that responded only, or the whole cohort)? If 42% of patients required dose reductions, what was the median tolerated dose (and did if vary among patients with a good vs poor ECOG PS)?
Author Response
There is a paucity of data on effective treatments for relapsed patients with systemic AL amyloidosis, especially for patients refractory to daratumumab. In this multicenter retrospective cohort study, Lebel et al. report the clinical outcomes of 26 heavily pretreated patients with relapsed system AL amyloidosis treated with venetoclax. Given the rarity of the disease, this is a sizeable cohort, and this paper adds to the evidence that venetoclax appears to be effective and relatively safe. The manuscript is thorough and very well written.
Minor feedback:
- Line 78-79: Please clarify how hematologic response was determined in the 31% of patients with concurrent MM?
Thank you for this important point. We clarified in the "methods" section: "For the purpose of this study, hematologic responses were documented similarly, per AL amyloidosis criteria, in patients who had concurrent MM".
- Line 86: what was the definition of “inadequate response” used for EFS?
Thank you for this important comment. In this context, "inadequate response" was not quantitively defined. Rather, it was defined qualitatively- if therapy was changed due to inadequate response according to the treating physician, an event was documented. Rather, a change in therapy due to other reasons (toxicity, patient preference or any other) was not considered an event for the EFS analysis.
- Line 109: the median age at venetoclax initiation does not match the first row I the table, please clarify
Thank you. We corrected.
- Table 1: Despite having 3.5 median prior lines of therapy, the median time from diagnosis to venetoclax initiation was 12 months, meaning that patients in this cohort had very short prior treatment durations – was this because of intolerance of lack of response? What was the best organ and hematologic response to prior therapy? (This will be helpful to determine if early organ responses to venetoclax may have been due to heme responses to prior therapy)
Unfortunately, we do not have available detailed data on all patients regarding heme and organ responses to previous therapies, as well as reasons for discontinuations of previous therapies. We documented the organ stage at diagnosis but also at the initiation of venetoclax therapy, when the disease was probably not in heme control in most cases. When we documented the organ responses, we considered the organ stage at the time of venetoclax initiation as the starting/reference point. Hence, we assume that any improvement at this point, from a specific organ state to a better state, can be related mostly to venetoclax. However, we agree that a component that related to previous periods of time when the light chains were under good control, cannot be excluded, as the organ response tends to be delayed in this disease. We added to the discussion: "That said, a delayed organ response that is related to previous therapies cannot be excluded, and cannot be differentiated accurately from the benefit that is attributed to venetoclax".
- Table 1: ANC at venetoclax initiation?
Thank you, we added
- Table 1: is the median prior therapies equivalent to the median prior lines?
Yes, but unfortunately there was a discrepancy between the text and the table, we corrected and clarified.
- Table 1: what was the median dFLC at initiation of venetoclax
Unfortunately, this data is not available. Responses were documented separately at each center and reported to the corresponding author without the specific LC values.
- Table 2: how can the lower range of time to any response and time to best response be zero? Wouldn’t that imply that patients responded on day zero of receiving venetoclax?
Thank you, the zero stem from rounding the correct number, we agree that this is very confusing and corrected. The earliest documented response was after 8 days, which is 0.26 months, rounded to 0.3m.
- Section 3.3: Of the 5 patients who died from disease complications, what were the causes of death?
3- sudden cardiac death, 1-heart failure, 1- unknown. These data were added to the manuscript.
- Line 168: In the EFS outcome, what proportion of events was due to organ/heme progression versus “inadequate response”
Thank you for this comment. All events were inadequate response or death.
- Line 182 and Table 2: Table 2 shows that the median “effective dose” was 400mg/day, how was this defined (is this the median dose among patients that responded only, or the whole cohort)? If 42% of patients required dose reductions, what was the median tolerated dose (and did if vary among patients with a good vs poor ECOG PS)?
By mentioning effective dose, we meant that we have taken into account drug interactions, which are common with venetoclax. For example, if a patient was on venetoclax 200mg without interacting medications, the dose is 200mg, but if this patient took an azole that is considered to double the effective dose, then we documented an effective dose of 400mg to this patient. This was not related to response, and in case that it is confusing we deleted the word "effective", and just left the comment regarding drug interactions.
The median tolerated dose was 400 mg (IQR 200-500). We added these data to the manuscript. The dose did not correlate with ECOG. Thank you.